# Synthesis of Thymidine Phosphorylase Inhibitor Based on Quinoxaline Derivatives and Their Molecular Docking Study

**DOI:** 10.3390/molecules24061002

**Published:** 2019-03-13

**Authors:** Noor Barak Almandil, Muhammad Taha, Rai Khalid Farooq, Amani Alhibshi, Mohamed Ibrahim, El Hassane Anouar, Mohammed Gollapalli, Fazal Rahim, Muhammad Nawaz, Syed Adnan Ali Shah, Qamar Uddin Ahmed, Zainul Amiruddin Zakaria

**Affiliations:** 1Department of Clinical Pharmacy, Institute for Research and Medical Consultations (IRMC), Imam Abdulrahman Bin Faisal University, P.O. Box 1982, Dammam 31441, Saudi Arabia; nbalmandil@iau.edu.sa (N.B.A.); msmibrahim@iau.edu.sa (M.I.); 2Department of Neuroscience Research, Institute for Research and Medical Consultations (IRMC), Imam Abdulrahman Bin Faisal University, P.O. Box 1982, Dammam 31441, Saudi Arabia; rkfarooq@iau.edu.sa (R.K.F.); ahalhibshi@iau.edu.sa (A.A.); 3Department of Chemistry, College of Sciences and Humanities, Prince Sattam bin Abdulaziz University, P.O. Box 83, Al-Kharij 11942, Saudi Arabia; e.anouar@psau.edu.sa; 4Department of Computer Information Systems, College of Computer Science & Information Technology, Imam Abdulrahman Bin Faisal University, P.O. Box 1982, Dammam 31441, Saudi Arabia; magollapalli@iau.edu.sa; 5Department of Chemistry, Hazara University, Mansehra 21300, Khyber Pakhtunkhwa, Pakistan; fazalstar@gmail.com; 6Department of Nano-Medicine Research, Institute for Research and Medical Consultations (IRMC), Imam Abdulrahman Bin Faisal University, P.O. Box 1982, Dammam 31441, Saudi Arabia; mnnmuhammad@iau.edu.sa; 7Faculty of Pharmacy, Universiti Teknologi MARA Puncak Alam Campus, 42300 Bandar Puncak Alam, Selangor D.E., Malaysia; benzene301@yahoo.com; 8Atta-ur-Rahman Institute for Natural Products Discovery (AuRIns), Universiti Teknologi MARA Puncak Alam Campus, 42300 Bandar Puncak Alam, Selangor D.E., Malaysia; 9Department of Pharmaceutical Chemistry, Kulliyyah of Pharmacy, International Islamic University Malaysia, 25200 Kuantan Pahang DM, Malaysia; quahmed@iium.edu.my; 10Department of Biomedical Science, Faculty of Medicine and Health Sciences, Universiti Putra Malaysia, 43400 Serdang, Selangor, Malaysia; 11Halal Institute Research Institute, Universiti Putra Malaysia, 43400 Serdang, Selangor, Malaysia

**Keywords:** quinoxaline analogs, synthesis, thymidine phosphorylase inhibition, molecular docking

## Abstract

We have synthesized quinoxaline analogs (**1**–**25**), characterized by ^1^H-NMR and HREI-MS and evaluated for thymidine phosphorylase inhibition. Among the series, nineteen analogs showed better inhibition when compared with the standard inhibitor 7-Deazaxanthine (IC_50_ = 38.68 ± 4.42 µM). The most potent compound among the series is analog **25** with IC_50_ value 3.20 ± 0.10 µM. Sixteen analogs **1**, **2**, **3**, **4**, **5**, **6**, **7**, **12**, **13**, **14**, **15**, **16**, **17**, **18**, **21** and **24** showed outstanding inhibition which is many folds better than the standard 7-Deazaxanthine. Two analogs **8** and **9** showed moderate inhibition. A structure-activity relationship has been established mainly based upon the substitution pattern on the phenyl ring. The binding interactions of the active compounds were confirmed through molecular docking studies.

## 1. Introduction:

Thymidine phosphorylase (TP), an enzyme involved in catabolism, exists in both prokaryotic and eukaryotic organisms [1,2,3]. TP speeds up the initial step in catabolism and converts thymidine nucleoside into thymine and 2-deoxy-d-ribose-1-phosphate by cleaving glycoside bond [4,5]. The intermediate obtained through dephosphorylation is 2-deoxy-d-ribose, which plays a significant role in prompting the tumor angiogenesis and hence favors cancer metastasis [6,7,8]. With respect to tumor angiogenesis, TP plays a major role, in that it helps in the proliferation process of endothelial cells throughout the body in cancer metastasis [9,10]. TP performs the same function as platelet endothelial cell growth factor (PD-ECGF) [11,12]. TP belonging to mammalians shares 39% sequence similarity with TP of *E. coli*, while the enzyme of mammalians also shares 65% resemblance with the active sites of residues of *E. coli* enzyme [13]. The production of 2′-deoxy-d-ribose can be limited through TP inhibitors which in turn suffocate the growth of tumor cells [14,15]. Therefore, medicinal chemists have tried to synthesize novel inhibitors of thymidine phosphorylase which have the potential to overcome the formation of new blood vessels and arrest the growth of tumor cells. Various attempts have been made to developed TP inhibitors [16,17,18,19,20,21,22,23]. The most potent inhibitor belonging to human TP known up to now is 5-chloro-6-[1-(2-iminopyrrolidinyl)methyl] uracil hydrochloride (TPI), while 7-deazaxanthine (7DX) is the first purine analog labeled as a TP inhibitor [24,25,26]. 

Nitrogen-containing heterocycles have attracted considerable attention due to their wide range of pharmacological importance [27,28]. Quinoxaline has a six-membered cyclic ring with two nitrogen atoms inside the cyclic ring. Quinoxaline and their analogs have attracted medicinal chemists over the decades and are used as antimicrobial [29], antibacterial [30], antifungal [31,32], anti-protozoan [33], anti-inflammatory, antianalgesic [34], anti-cancer [35,36], antidiabetic, and anti-proliferative agents [37,38]. Our research group has been working on the design and synthesis of heterocyclic compounds in search of potential lead compounds for many years and has found promising results [39,40,41,42,43,44,45,46,47,48,49]. 

In the past, several derivatives having six-member ring with two nitrogen reported to showed excellent inhibition of TP such as (**a**) to (**f**) in Figure 1 [9]. They showed outstanding activity which induced us to synthesize compounds having similar type of structure with low cast synthesis and simple chemistry to make synthesis adaptable for large scale synthesis. We report in this study new derivatives of quinoxalines with fused triazole and thiadiazole ring VII. The structure of our compounds is very close to the standard drug Deazaxanthine but our compounds have fused triazole and thiadiazole ring as well, which show much better activity than the standard.

## 2. Results and Discussion

### 2.1. Chemistry

Synthesis of quinoxaline derivatives (**1**–**25**) started with treating quinoxaline-2-carbohydrazide (**I**) with potassium thiocyanate in the presence of acid to form quinoxaline thiosemicarbazone (**II**) which was treated with a basic solution to cyclize and form 5-(quinoxalin-3-yl)-4H-1,2,4-triazole-3-thiol (**III**) which was treated with different substituted phenacyl bromide to afford (**1**–**25**) target compounds. The crude product was washed with water and recrystallized in methanol to afford pure product in 80–75%. All synthesized compounds (Scheme 1) were characterized by different spectroscopic methods (see Appendix A for full structures with activities).

### 2.2. In vitro Thymidine Phosphorylase Inhibitory Activity

We have synthesized 25 analogs of 5-phenyl-3-quinoxalin (**1**–**25**) and screened for inhibitory potential against thymidine phosphorylase enzyme. With respect to inhibitory potential, many analogs of the series showed a variable degree of inhibition with IC_50_ values ranging between 3.50 ± 0.20 to 56.40 ± 1.20 μM when compared with standard 7-Deazaxanthine (IC_50_ = 38.68 ± 1.12 μM). The analogs **1**, **2**, **3**, **4**, **5**, **6**, **7**, **12**, **13**, **14**, **15**, **16**, **17**, **18**, **21**, **24**, and **25** showed excellent inhibitory potential with IC_50_ values 13.60 ± 0.4, 26.10 ± 0.70, 18.10 ± 0.50, 27.40 ± 0.60, 33.40 ± 0.80, 24.40 ± 0.60, 34.70 ± 0.80, 33.20 ± 0.75, 18.30 ± 0.55, 13.20 ± 0.40, 15.20 ± 0.50, 3.50 ± 0.20, 24.20 ± 0.70, 16.90 ± 0.60, 26.20 ± 0.50, 13.10 ± 0.30 and 3.20 ± 0.10 μM respectively by comparing with standard 7-Deazaxanthine. Two analogs **8** and **9** showed moderate inhibitory activity with IC_50_ values 47.50 ± 0.90 and 56.40 ± 1.20 μM respectively, while six analogs **10, 11, 19, 20, 22**, and **23** were found inactive. Structure activity relationship has been established for all compounds, mainly based on substituents pattern of phenyl ring.

Compound **25**, a 2,3-dihydroxy analog was found to be the most active analog among the series with IC_50_ value 3.20 ± 0.10 μM. When comparing analog **25** with other dihydroxy analogs like **14**, a 2,4-dihydroxy analog (IC_50_ = 13.20 ± 0.40 μM) **15,** a 2,5-dihydroxy analog (IC_50_ = 15.20 ± 0.50 μM) and **16,** a 2,4-dihydroxy analog (IC_50_ = 3.50 ± 0.20 μM), analog **25** was found to be superior. Although all the four analogs have two hydroxyl groups at the phenyl ring, the position of attachment on phenyl ring are different. The difference in inhibitory activity of these four analogs seems due to the different position of the hydroxyl group on the phenyl ring, as seen in Figure 2.

When comparing dihydroxy analogs with monohydroxy analog like **12, 13, 17, 18, 21**, and **24** the dihydroxy analogs were found to be more potent. This greater potential seems to be due to the greater number of hydroxy groups on the phenyl ring.

Similarly, a pattern was also observed in flourine substituted analogs like **1, 2**, and **3** IC_50_ value 13.60 ± 0.4, 26.10 ± 0.70 and 18.10 ± 0.50 μM respectively. All three analogs possess flouro group at the phenyl ring, but the analog **1** shows greater potential than analogs **2** and **3**. The difference in the inhibitory potential in analog **1, 2** and **3** seems due to attachment of flouro group at various positions on the phenyl part, as seen in Figure 3.

The same trend of difference in inhibitory activity was found in chloro substituted analogs **4, 5** and **6** with IC_50_ value 27.40 ± 0.60, 33.40 ± 0.80 and 24.40 ± 0.60 μM respectively. All three analogs have chloro group but their attachment on phenyl ring differs from each other, and the difference in inhibitory activity seems to be due to the attachment of chloro group at variable positions on the phenyl part Figure 4.

So, it was concluded that in our designed molecules the position, nature, and number of substituent play critical role in thymidine phosphorylase inhibition.

### 2.3. Molecular Docking

The IC_50_ values of quinoxaline derivatives as thymidine phosphorylase inhibitors are shown in Table 1. The thymidine phosphorylase inhibition by the synthesized derivatives is mainly due to the type, number, and positions of the functional group in the substitute group R of the basic skeleton (Table 1). For a better understanding of the enzyme inhibition by the synthesized compounds, molecular docking study has been carried out to shed light on the established binding modes of the four selected synthesized compounds **14**, **15**, **16**, and **25**. The selected compounds differ by the substitution position of the hydroxyl group in the aromatic ring (Table 1). Compounds **16** and **25** with OH groups in *meta* position to each other show higher activity than **14** and **15** where OH groups are in *para* position to each other (Table 1). Table 2 summarizes the calculated binding energies of the stable complex’s ligand thymidine phosphorylase, number of established intermolecular hydrogen bonding between the synthesized compounds (**14**, **15**, **16** and **25**) and active site residues of thymidine phosphorylase.

The complexes formed between the docked selected compounds and amino acids of the binding active site of thymidine phosphorylase exhibited negative binding energies, which is a signpost that the inhibition of thymidine phosphorylase by the selected compounds is thermodynamically favorable (Table 2). As can be seen from the docking results in Table 2 and Figure 5 and Figure 6, the highest activity of **16** and **25** compared with **14** and **15** mainly refers to the stability of the formed complexes between the docked compounds (**16** and **25**) and thymidine phosphorylase compared with the formed complexes between the docked compounds (**14** and **15**) and thymidine phosphorylase ones. The higher activity of **25** compared with **16** may refer to the number of intermolecular hydrogen bonding formed with substituted OH groups in the complex **25**-receptor compared to **16**-receptor one. Indeed, three hydrogen bonds are formed between OH groups of **25** and SER 86 and HIS 85 of the active site of thymidine phosphorylase of 1.71, 2.24, and 3.38 Å, respectively. However, two hydrogen bonds are formed between OH groups of **16** and THR 120 of the active site of thymidine phosphorylase of 1.26 and 1.78 Å, respectively.

## 3. Experimental Section

### 3.1. General Methods

All nuclear magnetic resonance experiments were carried out using on Avance Bruker 500 MHz (Wissembourg, Switzerland)). Electron impact mass spectra (EI–MS) were recorded on a Finnigan MAT-311A (Bremen, Germany). Thin layer chromatography (TLC) was performed on pre-coated silica gel aluminum plates (Kieselgel 60, 254, E. Merck, Darmstadt, Germany). Chromatograms were visualized by UV at 254 and 365 nm.

#### 3.1.1. Thymidine Phosphorylase Assay

Since human TP is not easy to obtain, we used commercially available recombinant *E. coli* TP. The primary sequence of TP is frequently preserved throughout evolution as mammalian TP is reported to share 39% sequence resemblance with the TP of *E. coli*. The mammalian enzyme also shared up to 70% resemblance with the active site residues, and three-dimensional structure of *E. coli* TP enzyme [49]. The thymidine phosphorylase/PD-ECGF (*E. coli*) activity was determined by measuring the absorbance at 290 nm spectrophotometrically. The method was described in [50,51]. In brief, the total reaction mixture of 200 µL contained 145 µL of potassium phosphate buffer (pH 7.4), 30 µL of enzyme (human and *E. coli*) at concentration 0.05 and 0.002 U, respectively, were incubated with 5 µL of test materials for 10 min at 25 °C in a microplate reader. After incubation, a pre-read at 290 nm was taken to deduce the absorbance of substrate particles. The substrate (20 µL, 1.5 mM), dissolved in potassium phosphate buffer, was immediately added to the plate and continuously read after 10, 20, and 30 min in a microplate reader (SpectraMax, Molecular Devices, CA, USA). All assays were performed in triplicate.

#### 3.1.2. Calculations

Reactions for above mentioned biological activities were carried out in triplicate. Results were then processed using SoftMax Pro 4.8 software (Molecular Devices, San Jose, CA, USA) and then by Microsoft Excel. The percent inhibition for above mentioned biological activities was calculated by following formula:Percent Inhibition = 100 − (ODtest compound/ODcontrol) × 100(1)

#### 3.1.3. Synthesis of Quinoxaline Thiosemicarbazone (II)

Quinoxaline-2-carbohydrazide (5 g, 26.60 mmol), potassium thiocyanate (2.61 g, 26.90 mmol), and 4 mL of conc. HCl in 40 mL of water were refluxed for 4 h. The reaction progress was monitored by TLC. After completion of the reaction, the reaction mixture was left for cooling and white solid ppts appeared then the solid was filtered and dried. Yield 5.62 g (85.6%); m.p. 289–290 °C.

#### 3.1.4. Synthesis of 5-(quinoxalin-3-yl)-4H-1,2,4-triazole-3-thiol (III)

The 5-(quinoxalin-3-yl)-4H-1,2,4-triazole-3-thiol (**III**) was synthesized as reported in [52].

### 3.2. General Procedure for Synthesis of Quinoxaline Derivatives (***1**–**25***)

5-(quinoxalin-3-yl)-4H-1,2,4-triazole-3-thiol (**III**) (1 mmol) was refluxed with appropriate arylacyl bromide (1 mmol) in 15 mL ethanol for 12 h. The reaction was monitored by TLC. After completion of the reaction, the product was left for cooling. The solid was filtered and the crude products were recrystallized from methanol (all synthesized compounds with their SMILE structures and activities are provided in Appendix A).

#### 3.2.1. 5-(2-flourophenyl)-3(Quinoxalin-2yl)thiazolo[2,3-c][1,2,4]triazole

Yield: 81%. m.p.: 299–300 °C; ^1^H-NMR (500 MHz, DMSO-*d_6_*): δ 8.32 (s, 1H), 8.02 (d, *J* = 8.0 Hz, 1H), 7.88 (d, *J* = 7.0 Hz, 1H), 7.70–7.67 (m, 3H), 7.52–7.48 (m, 2H), 7.30–7.26 (m, 2H). ^13^C-NMR (150 MHz, DMSO -*d_6_*): δ 158.5, 155.5, 145.9, 145.4, 144.5, 142.4, 142.3, 141.11, 130.5, 129.8, 129.7, 129.5, 129.4, 129.3, 124.10, 123.7, 115.7, 114.9. HR-ESI-MS: *m/z* calcd for C_18_H_10_FN_5_S, [M]^+^ 347.0641; Found 347.0623.

#### 3.2.2. 5-(3-flourophenyl)-3(Quinoxalin-2yl)thiazolo[2,3-c][1,2,4]triazole

Yield: 80%. m.p.: 304–305 °C. ^1^H-NMR (500 MHz, DMSO-*d_6_*): δ 8.10 (s, 1H), 7.90 (d, *J* = 7.5 Hz, 1H), 7.67 (d, *J* = 7.5 Hz, 2H), 7.61 (d, *J* = 8.0 Hz, 1H), 7.56 (d, *J* = 7.5 Hz, 1H), 7.50–7.46 (m, 2H), 7.22 (t, *J* = 8.0 Hz, 1H), 7.18–7.16 (m, 1H); ^13^C-NMR (150 MHz, DMSO -*d_6_*): δ 162.2, 155.5, 145.9, 145.4, 144.5, 142.4, 142.3, 141.11, 134.8, 129.8, 129.7, 129.5, 129.4, 127.7, 115.7, 123.3, 115.10, 115.7. HR-ESI-MS: *m/z* calcd for C_18_H_10_FN_5_S, [M]^+^ 347.0641; Found 347.0625.

#### 3.2.3. 5-(4-flourophenyl)-3(Quinoxalin-2yl)thiazolo[2,3-c][1,2,4]triazole

Yield: 77%. m.p.: 308–309 °C. ^1^H-NMR (500 MHz, DMSO-*d_6_*): δ 8.11(s, 1H), 7.90 (d, *J* = 7.5 Hz, 1H), 7.80 (d, *J* = 8.0 Hz, 2H), 7.72–7.68 (m, 2H), 7.50–7.45 (m, 2H), 7.24 (d, *J* = 7.5 Hz, 2H); ^13^C-NMR (150 MHz, DMSO -*d_6_*): δ 162.9, 155.5, 145.9, 145.5, 144.5, 142.4, 142.3, 141.11, 130.8, 130.7, 129.8, 129.7, 129.5, 129.4, 128.8, 116.3, 116.2, 115.7. HR-ESI-MS: *m/z* calcd for C_18_H_10_FN_5_S, [M]^+^ 347.0641; Found 347.0617.

#### 3.2.4. 5-(2-chlorophenyl)-3(Quinoxalin-2yl)thiazolo[2,3-c][1,2,4]triazole

Yield: 82%. m.p.: 280–281 °C. ^1^H-NMR (500 MHz, DMSO-*d_6_*): δ 8.50 (s, 1H), 8.12 (d, *J* = 7.0 Hz, 1H), 7.90 (d, *J* = 7.0 Hz, 1H), 7.72–7.68 (m, 3H), 7.54–7.49 (m, 2H), 7.42–7.38 (m, 2H); ^13^C-NMR (150 MHz, DMSO -*d_6_*): δ 155.5, 145.9, 145.4, 144.5, 142.4, 142.3, 141.11, 132.7, 132.4, 130.8, 130.3, 129.9, 129.8, 129.7, 129.5, 129.4, 128.9, 115.7. HR-ESI-MS: *m/z* calcd for C_18_H_10_ClN_5_S, [M]^+^ 363.0345; Found 363.0319.

#### 3.2.5. 5-(3-chlorophenyl)-3(Quinoxalin-2yl)thiazolo[2,3-c][1,2,4]triazole

Yield: 80%. m.p.: 285–286 °C. ^1^H-NMR (500 MHz, DMSO-*d_6_*): δ 8.90 (s, 1H), 8.57 (d, *J* = 6.5 Hz, 1H), 8.16 (d, *J* = 7.5 Hz, 1H), 8.05 (s, 1H), 7.86 (d, *J* = 8.0 Hz, 1H), 7.70 (d, *J* = 7.5 Hz, 2H), 7.50–7.47 (m, 2H), 7.42 (d, *J* = 8.0 Hz, 1H); ^13^C-NMR (150 MHz, DMSO -*d_6_*): δ 155.5, 145.9, 145.4, 144.5, 142.4, 142.3, 141.11, 134.9, 134.6, 129.9, 129.8, 129.7, 129.6, 129.5, 129.4, 128.9, 125.8, 115.7. HR-ESI-MS: *m/z* calcd for C_18_H_10_ClN_5_S, [M]^+^ 363.0345; Found 363.0323.

#### 3.2.6. 5-(4-chlorophenyl)-3(Quinoxalin-2yl)thiazolo[2,3-c][1,2,4]triazole

Yield: 89%. m.p.: 249–250 °C. 1H-NMR (500 MHz, DMSO-d6): δ 8.08 (s, 1H), 7.88 (d, *J* = 7.5 Hz, 1H), 7.70 (d, *J* = 8.5 Hz, 2H), 7.69–7.64 (m, 2H), 7.52–7.45 (m, 4H). ^13^C-NMR (150 MHz, DMSO -*d_6_*): δ 155.5, 145.9, 145.4, 144.5, 142.4, 142.3, 141.11, 134.5, 131.3, 129.9, 129.8, 129.7, 129.6, 129.5, 129.4, 128.9, 127.9, 115.7. HR-ESI-MS: *m/z* calcd for C_18_H_10_ClN_5_S, [M]^+^ 363.0345; Found 363.0327.

#### 3.2.7. 5-(2-nitrophenyl)-3(Quinoxalin-2yl)thiazolo[2,3-c][1,2,4]triazole

Yield: 83%. m.p.: 310–311 °C. ^1^H-NMR (500 MHz, DMSO-*d_6_*): δ 8.26 (d, *J* = 7.0 Hz, 1H), 7.98 (d, *J* = 7.5 Hz, 1H), 7.90 (d, *J* = 7.5 Hz, 1H), 7.74 (d, *J* = 7.0 Hz, 1H), 7.70 (d, *J* = 7.0 Hz, 1H), 7.63 (d, *J* = 7.0 Hz, 1H), 7.61 (d, *J* = 7.0 Hz, 1H), 7.50–7.46 (m, 2H), 7.43 (s, 1H); ^13^C-NMR (150 MHz, DMSO -*d_6_*): δ 155.5, 148.9, 145.9, 145.5, 144.5, 142.4, 142.3, 141.11, 135.5, 132.8, 129.9, 129.8, 129.7, 129.5, 129.4, 125.4, 124.6,115.7. HR-ESI-MS: *m/z* calcd for C_18_H_10_N_6_O_2_S, [M]^+^ 374.0586; Found 374.0568.

#### 3.2.8. 5-(4-nitrophenyl)-3(Quinoxalin-2yl)thiazolo[2,3-c][1,2,4]triazole

Yield: 81%. m.p.: 315–316 °C. ^1^H-NMR (500 MHz, DMSO-*d_6_*): δ 8.22 (d, *J* = 8.3 Hz, 2H), 8.18 (s, 1H), 7.98 (d, *J* = 8.0 Hz, 2H), 7.90 (d, *J* = 7.5 Hz, 1H), 7.70–7.65 (m, 2H), 7.54 (d, *J* = 7.0 Hz, 1H), 7.44 (s, 1H); ^13^C-NMR (150 MHz, DMSO -*d_6_*): δ 155.5, 147.9, 145.9, 145.4, 144.5, 142.4, 142.3, 142.1, 141.9, 141.8, 139.3, 129.8, 129.7, 129.5, 129.4, 126.4, 126.3, 124.6,124.5,115.7. HR-ESI-MS: *m/z* calcd for C_18_H_10_N_6_O_2_S, [M]^+^ 374.0586; Found 374.0559.

#### 3.2.9. 3-(quinoxalin-2-yl)-5-o-tolylthiazolo[2,3-c][1,2,4]triazole

Yield: 88%. m.p.: 265-266 °C. ^1^H-NMR (500 MHz, DMSO-*d_6_*): δ 8.40 (s, 1H), 7.89 (d, *J* = 8.0 Hz, 2H), 7.68 (d, *J* = 7.0 Hz, 2H), 7.48–7.42 (m, 2H), 7.35–7.22 (m, 3H), 2.49 (s, 3H); ^13^C-NMR (150 MHz, DMSO -*d_6_*): *δ* 155.5, 145.9, 145.4, 144.5, 142.4, 142.3, 141.11, 136.9, 130.1, 129.9, 129.8, 129.7, 129.5, 129.4, 128.8, 126.4, 122.9, 115.7, 18.9. HR-ESI-MS: *m/z* calcd for C_19_H_13_N_5_S, [M]^+^ 343.0892; Found 343.0864.

#### 3.2.10. 3-(quinoxalin-2-yl)-5-m-tolylthiazolo[2,3-c][1,2,4]triazole

Yield: 84%. M.P.: 270–271 °C. ^1^H-NMR (500 MHz, DMSO-*d_6_*): δ 8.08 (s, 1H), 7.90 (d, *J* = 8.0 Hz, 1H), 7.72–7.68 (m, 2H), 7.57 (s, 1H), 7.52 (d, *J* = 7.0 Hz, 1H), 7.43 (s, 1H), 2.46 (s, 3H); ^13^C-NMR (150 MHz, DMSO -*d_6_*): δ 155.5, 145.9, 145.4, 144.5, 142.4, 142.3, 141.11, 139.1, 133.1, 130.6, 129.9, 129.8, 129.7, 129.5, 129.4, 128.9, 124.6, 115.7, 21.8. HR-ESI-MS: *m/z* calcd for C_19_H_13_N_5_S, [M]^+^ 343.0892; Found 343.0871.

#### 3.2.11. 3-(quinoxalin-2-yl)-5-p-tolylthiazolo[2,3-c][1,2,4]triazole

Yield: 81%. m.p.: 280–281 °C. ^1^H-NMR (500 MHz, DMSO-*d_6_*): δ 8.10 (s, 1H), 7.88 (d, *J* = 8.0 Hz, 1H), 7.71–7.65 (m, 4H), 7.47–7.42 (m, 2H), 7.27 (d, *J* = 7.9 Hz, 2H), 2.49 (s, 3H); ^13^C-NMR (150 MHz, DMSO -*d_6_*): δ 155.5, 145.9, 145.4, 145.3, 144.5, 142.4, 142.3, 141.11, 130.2, 131.9, 129.9, 129.8, 129.7, 129.5, 129.4, 129.3, 125.9, 125.7, 115.7, 21.5. HR-ESI-MS: *m/z* calcd for C_19_H_13_N_5_S, [M]^+^ 343.0892; Found 343.0873.

#### 3.2.12. 3-(3-(quinoxalin-2-yl)thiazolo[2,3-c][1,2,4]triazol-5-yl)phenol

Yield: 85%. m.p.: 289–290 °C. ^1^H-NMR (500 MHz, DMSO-*d_6_*): δ 9.05 (s, 1H, OH), 8.04 (s, 1H), 7.90 (d, *J* = 7.5 Hz, 1H), 7.70–7.65 (m, 2H), 7.49–7.43 (m, 2H), 7.25 (d, *J* = 7.5 Hz, 1H), 7.19–7.15 (m, 1H), 7.15 (d, *J* = 7.0 Hz, 1H), 6.81 (d, *J* = 7.0 Hz, 1H); ^13^C-NMR (150 MHz, DMSO -*d_6_*): δ 157.7, 155.5, 145.9, 145.4, 144.5, 142.4, 142.3, 141.11, 134.6, 130.8, 129.8, 129.7, 129.5, 129.4, 120.3, 115.10, 115.9, 115.7. HR-ESI-MS: *m/z* calcd for C_18_H_11_N_5_OS, [M]^+^ 345.0684; Found 345.0666.

#### 3.2.13. 4-(3-(quinoxalin-2-yl)thiazolo[2,3-c][1,2,4]triazol-5-yl)phenol

Yield: 82%. m.p.: 297–298 °C. ^1^H-NMR (500 MHz, DMSO-*d_6_*): δ 9.82 (s, 1H, OH), 8.03 (s, 1H), 7.88 (d, *J* = 8.0 Hz, 1H), 7.65–7.62 (m, 2H), 7.58 (d, *J* = 7.5 Hz, 2H), 7.46–7.40 (m, 2H), 6.80 (d, *J* = 8.0 Hz, 2H). ^13^C-NMR (150 MHz, DMSO -*d_6_*): δ 158.7, 155.5, 145.9, 145.5, 144.5, 142.4, 142.3, 141.11, 129.8, 129.7, 129.5, 129.4, 128.9, 128.7, 125.7, 116.6, 116.4,115.7. HR-ESI-MS: *m/z* calcd for C_18_H_11_N_5_OS, [M]^+^ 345.0684; Found 345.0661.

#### 3.2.14. 4-(3-(quinoxalin-2-yl)thiazolo[2,3-c][1,2,4]triazol-5-yl)benzene-1,3-diol

Yield: 79%. m.p.: 299–300 °C. ^1^H-NMR (500 MHz, DMSO-*d_6_*): δ 10.70 (s, 1H, OH), 9.80 (s, 1H, OH), 8.18 (s, 1H), 7.86 (d, *J* = 8.0 Hz, 1H), 7.71–7.65 (m, 3H), 7.50–7.43 (m, 2H), 6.32 (d, *J* = 7.0, 6.0 Hz, 2H). ^13^C-NMR (150 MHz, DMSO -*d_6_*): δ 160.1, 156.2, 155.5, 145.9, 145.5, 144.5, 142.4, 142.3, 141.11, 133.4, 129.8, 129.7, 129.5, 129.4, 113.3, 109.0, 105.8, 115.7. HR-ESI-MS: *m/z* calcd for C_18_H_11_N_5_O_2_S, [M]^+^ 361.0633; Found 361.0615.

#### 3.2.15. 2-(3-(quinoxalin-2-yl)thiazolo[2,3-c][1,2,4]triazol-5-yl)benzene-1,4-diol

Yield: 87%. m.p.: 301–302 °C. ^1^H-NMR (500 MHz, DMSO-*d_6_*): δ 9.72 (s, 2H, OH), 8.30 (s, 1H), 7.88 (d, *J* = 8.0 Hz, 1H), 7.74–7.65 (m, 2H), 7.52–7.40 (m, 2H), 7.17 (d, *J* = 6.0 Hz, 1H), 6.75 (d, *J* = 8.0 Hz, 1H), 6.68 (d, *J* = 7.0 Hz, 1H); ^13^C-NMR (150 MHz, DMSO -*d_6_*): δ 155.5, 150.3, 147.9, 145.9, 145.5, 144.5, 142.4, 142.3, 141.11, 129.8, 129.7, 129.5, 129.4, 122.1, 117.9, 117.5, 115.7, 114.5. HR-ESI-MS: *m/z* calcd for C_18_H_11_N_5_O_2_S, [M]^+^ 361.0633; Found 361.0617.

#### 3.2.16. 4-(3-(quinoxalin-2-yl)thiazolo[2,3-c][1,2,4]triazol-5-yl)benzene-1,2-diol

Yield: 83%. m.p.: 293–294 °C. ^1^H-NMR (500 MHz, DMSO-*d_6_*): δ 9.30 (s, 1H, OH), 9.10 (s, 1H, OH), 7.96 (s, 1H), 7.82 (d, *J* = 8.0 Hz, 1H), 7.68(d, *J* = 6.0 Hz, 2H), 7.53–7.40 (m, 2H), 7.20 (s, 1H), 6.95 (d, *J* = 7.0 Hz, 1H), 6.70 (d, *J* = 8.0 Hz, 1H); ^13^C-NMR (150 MHz, DMSO -*d_6_*): δ 155.5, 147.5, 145.9, 145.7, 145.5, 144.5, 142.4, 142.3, 141.11, 129.8, 129.7, 129.5, 129.4, 127.2, 121.7, 116.4, 115.7, 114.5. HR-ESI-MS: *m/z* calcd for C_18_H_11_N_5_O_2_S, [M]^+^ 361.0633; Found 361.0612.

#### 3.2.17. 5-methoxy-2-(3-(quinoxalin-2-yl)thiazolo[2,3-c][1,2,4]triazol-5-yl)phenol

Yield: 86%. m.p.: 305–306 °C. ^1^H-NMR (500 MHz, DMSO-*d_6_*): δ 10.87 (s, 1H, OH), 8.30 (s, 1H), 7.82 (d, *J* = 7.5 Hz, 1H), 7.62 (s, 2H), 7.55 (d, *J* = 8.0 Hz, 1H), 7.46–7.40 (s, 2H), 6.51 (d, *J* = 8.0 Hz, 1H), 6.42 (d, *J* = 6.0 Hz, 1H), 3.75 (s, 3H); ^13^C-NMR (150 MHz, DMSO -*d_6_*): δ 162.2, 156.2, 155.5, 145.9, 145.5, 144.5, 142.4, 142.3, 141.11, 132.9, 129.8, 129.7, 129.5, 129.4, 115.7, 112.9, 107.6, 104.4, 55.9. HR-ESI-MS: *m/z* calcd for C_19_H_13_N_5_O_2_S, [M]^+^ 375.0790; Found 375.0772.

#### 3.2.18. 2-methoxy-5-(3-(quinoxalin-2-yl)thiazolo[2,3-c][1,2,4]triazol-5-yl)phenol

Yield: 83%. m.p.: 293–294 °C. ^1^H-NMR (500 MHz, DMSO-*d_6_*): δ 9.22 (s, 1H, OH), 8.04 (s, 1H), 7.88 (d, *J* = 8.0 Hz, 1H), 7.76–7.69 (m, 2H), 7.46–7.41 (m, 2H), 7.28 (d, *J* = 6.0Hz, 1H), 7.04 (d, *J* = 8.0 Hz, 1H), 6.94 (d, *J* = 8.0 Hz, 1H), 3.77 (s, 3H); ^13^C-NMR (150 MHz, DMSO -*d_6_*): δ 155.5, 145.7, 147.5, 147.4, 145.5, 144.5, 142.4, 142.3, 141.11, 129.8, 129.7, 129.5, 129.4, 126.9, 121.7, 114.1, 115.7, 111.6, 56.3. HR-ESI-MS: *m/z* calcd for C_19_H_13_N_5_O_2_S, [M]^+^ 375.0790; Found 375.0768.

#### 3.2.19. 5-(3-methoxyphenyl)-3-(quinoxalin-2-yl)thiazolo[2,3-c][1,2,4]triazole

Yield: 75%. m.p.: 179–180 °C. ^1^H-NMR (500 MHz, DMSO-*d_6_*): δ 8.10 (s, 1H), 7.90 (d, *J* = 8.0 Hz, 1H), 7.70–7.59 (m, 3H), 7.46–7.40 (m, 2H), 7.30 (d, *J* = 6.5 Hz, 2H), 6.96–6.89 (m, 1H), 3.80 (s, 3H); ^13^C-NMR (150 MHz, DMSO -*d_6_*): δ 161.2, 155.5, 145.7, 145.5, 144.5, 142.4, 142.3, 141.11, 134.2, 130.3, 129.8, 129.7, 129.5, 129.4, 119.9, 115.7, 114.1, 113.8, 55.9. HR-ESI-MS: *m/z* calcd for C_19_H_13_N_5_OS, [M]^+^ 359.0841; Found 359.0843.

#### 3.2.20. 5-(4-methoxyphenyl)-3-(quinoxalin-2-yl)thiazolo[2,3-c][1,2,4]triazole

Yield: 87%. m.p.: 301–302 °C. ^1^H-NMR (500 MHz, DMSO-*d_6_*): δ 8.07 (s, 1H), 7.88 (d, *J* = 8.0 Hz, 2H), 7.70 (d, *J* = 8.0 Hz, 2H), 7.62 (d, *J* = 7.0 Hz, 1H), 7.47–7.42 (s, 2H), 7.02 (d, *J* = 8.0 Hz, 2H), 3.81 (s, 3H); ^13^C-NMR (150 MHz, DMSO -*d_6_*): δ 160.8, 155.5, 145.7, 145.5, 144.5, 142.4, 142.3, 141.11, 129.8, 129.7, 129.5, 129.4, 128.7, 128.5, 125.5, 115.7, 114.9, 114.7, 55.9. HR-ESI-MS: *m/z* calcd for C_19_H_13_N_5_OS, [M]^+^ 359.0841; Found 359.0819.

#### 3.2.21. 4-methoxy-2-(3-(quinoxalin-2-yl)thiazolo[2,3-c][1,2,4]triazol-5-yl)phenol

Yield: 83%. m.p.: 308–309 °C. ^1^H-NMR (500 MHz, DMSO-*d_6_*): δ 9.98 (s, 1H, OH), 8.40 (s, 1H), 7.88 (d, *J* = 8.2 Hz, 1H), 7.72–7.67 (m, 2H), 7.47–7.42 (m, 2H), 7.30 (s, 1H), 6.82 (d, *J* = 2.0 Hz, 2H), 3.79 (s, 3H); ^13^C-NMR (150 MHz, DMSO -*d_6_*): δ 155.5, 153.9, 145.7, 147.5, 145.4, 144.5, 142.4, 142.3, 141.11, 129.8, 129.7, 129.5, 129.4, 121.7, 117.6, 115.7, 115.5, 112.9, 55.9. HR-ESI-MS: *m/z* calcd for C_19_H_13_N_5_O_2_S, [M]^+^ 375.0790; Found 375.0772.

#### 3.2.22. 5-(pyridin-3-yl)-3-(quinoxalin-2-yl)thiazolo[2,3-c][1,2,4]triazole

Yield: 82%. m.p.: 276–277 °C. ^1^H-NMR (500 MHz, DMSO-*d_6_*): δ 8.42 (s, 1H), 8.12 (d, *J* = 8.0 Hz, 1H), 7.90 (d, *J* = 7.0 Hz, 1H), 7.70–7.65 (m, 4H), 7.51 (d, *J* = 7.0 Hz, 1H), 7.47–7.42 (m, 2H). ^13^C-NMR (150 MHz, DMSO -*d_6_*): δ 147.9, 145.7, 147.5, 145.4, 144.5, 143.4, 142.4, 142.3, 141.11, 134.2, 133.2, 129.8, 129.7, 129.5, 129.4, 124.2,120.3. HR-ESI-MS: *m/z* calcd for C_17_H_10_N_6_S, [M]^+^ 330.0688; Found 330.0667.

#### 3.2.23. 5-(pyridin-4-yl)-3-(quinoxalin-2-yl)thiazolo[2,3-c][1,2,4]triazole

Yield: 86%. m.p.: 276–277 °C. ^1^H-NMR (500 MHz, DMSO-*d_6_*): δ 8.60 (d, *J* = 7.0 Hz, 2H), 8.13 (s, 1H), 7.92 (d, *J* = 7.9 Hz, 1H), 7.74–7.66 (m, 4H), 7.50–7.42 (m, 2H). ^13^C-NMR (150 MHz, DMSO -*d_6_*): δ 149.9, 149.8, 145.7, 145.5, 144.5, 143.4, 142.4, 142.3, 141.11, 140.5, 129.8, 129.7, 129.5, 129.4, 121.5, 121.5, 120.3. HR-ESI-MS: *m/z* calcd for C_17_H_10_N_6_S, [M]^+^ 330.0688; Found 330.0670.

#### 3.2.24. 2-(3-(quinoxalin-2-yl)thiazolo[2,3-c][1,2,4]triazol-5-yl)phenol

Yield: 82%. m.p.: 286–287 °C. ^1^H-NMR (500 MHz, DMSO-*d_6_*): δ 10.54 (s, 1H, OH), 8.20 (s, 1H), 7.90 (d, *J* = 7.5 Hz, 1H), 7.72 (d, *J* = 7.0 Hz, 1H), 7.65 (d, *J* = 8.0 Hz, 2H), 7.52–7.44 (m, 2H), 7.25–7.20 (m, 2H), 6.90 (d, *J* = 6.5 Hz, 1H). ^13^C-NMR (150 MHz, DMSO -*d_6_*): δ 155.5, 155.3, 145.7, 145.5, 144.5, 142.4, 142.3, 141.11, 131.7, 130.3, 129.8, 129.7, 129.5, 129.4, 120.7, 117.9, 121.7, 115.7. HR-ESI-MS: *m/z* calcd for C_18_H_11_N_5_OS, [M]^+^ 345.0684; Found 345.0668.

#### 3.2.25. 3-(3-(quinoxalin-2-yl)thiazolo[2,3-c][1,2,4]triazol-5-yl)benzene-1,2-diol

Yield: 82%. m.p.: 309–310 °C. ^1^H-NMR (500 MHz, DMSO-*d_6_*): δ 10.08 (s, 1H, OH), 9.35 (s, 1H, OH), 8.40 (s, 1H), 7.88 (d, *J* = 8.0 Hz, 1H), 7.64 (d, *J* = 6.0 Hz, 2H), 7.45–7.40 (m, 2H), 7.14 (d, *J* = 7.0 Hz, 1H), 6.80 (d, *J* = 7.0 Hz, 1H), 6.70 (t, *J* = 7.5 Hz, 1H); ^13^C-NMR (150 MHz, DMSO -*d_6_*): δ 155.5, 145.7, 145.5, 145.4, 144.5, 143.7, 142.4, 142.3, 141.11, 129.8, 129.7, 129.5, 129.4, 124.3, 123.4, 122.1, 117.5, 115.7. HR-ESI-MS: *m/z* calcd for C_18_H_11_N_5_O_2_S, [M]^+^ 361.0633; Found 361.0605.

### 3.3. Molecular Docking Details

The intermolecular binding modes between the docked selected synthesized quinoxaline derivatives and the active residues of thymidine phosphorylase have been explored using AutoDock package (The Scripps Research Institute, La Jolla, CA, USA) [53]. The geometries of thymidine phosphorylase and the original docked ligand 3′-azido-2′-fluoro-dideoxyuridine were obtained from the Research Collaboratory for Structural Bioinformatics (RCSB) data bank website (PDB code 4EAD) [54]. Water molecules were removed; polar hydrogen atoms and Kollman charge were added to the extracted receptor using the automated tool in AutoDock Tools 4.2. The active site is identified based on the co-crystallized receptor–ligand complex structure of thymidine phosphorylase. The re-docking of the original ligand 3′-azido-2′-fluoro-dideoxyuridine into the active site is well reproduced with an RMSD value less than 1.14 Å and a binding energy of −6.63 kcal/mol. The molecular structures geometries of quinoxaline derivatives were minimized at Merck molecular force field 94 (MMFF94) level44. The optimized structures were saved as PDB files. Nonpolar hydrogens were merged, and rotatable bonds were defined for each docked ligand. Docking studies were performed by the Lamarckian genetic algorithm, with 500 as a total number of the run for binding site for original ligand the synthesized derivatives. In each set, a population of 150 individuals with 27,000 generations and 250,000 energy evaluations were employed. Operator weights for crossover, mutation, and elitism were set to 0.8, 0.02, and 1, respectively. The binding site was defined using a grid of 35 × 35 × 35 points each with a grid spacing of 0.375 Å. The docking calculation has been carried out using an Intel (R) Core (TM) i5-3770 CPU @ 3.40 GHz workstation.

## 4. Conclusions

Twenty-five quinoxaline analogs (**1**–**25**) were synthesized. All synthesized compounds were reported first having novel structures. They were screened against thymidine phosphorylase. The result profile showed that dihydroxyl substituted compounds showed excellent activity along with some variation depending upon their position. The halogen groups, on other hand, showed a significant inhibitory potential against thymidine phosphorylase. The fluorine substituents showed better activity than the chlorines. The binding interactions of the most active analogs were determined by molecular docking study. That confirms the binding interactions of active compounds with enzyme.

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
