# Peer review of "Synthesis of Thymidine Phosphorylase Inhibitor Based on Quinoxaline Derivatives and Their Molecular Docking Study"

_molecules, 2019, doi:10.3390/molecules24061002_

Round 1
Reviewer 1 Report
I don't have any particular commments or suggestions for you. Your manuscript is worth being published after these following little amendments:
On page 2, line 32 replace "analog" with "compound"
On page 3, line 58 replace "belong" with "belonging"
On page 5, lines 81 and 82, I can't get why you mention the "scheme 1" in the sentence "All synthesized compounds were characterized by different spectroscopic methods given in scheme 1". As a matter of fact, scheme 1 just shows the synthesis of derivatives 1 - 25 and no different spectroscopic methods..
On page 9, line 117, replace "flouro" with "fluoro"
On page 10, line 137, replace "five" with "four"
On page 10, line 139, replace "(Scheme 1)" with "(Table 1)"
On page 11, line 154, replace "bending" with "binding"
On page 11, line 161, add "of" between "number" and "intermolecular"
On page 12, line 169 you have to replace "Figure 1: 3D (right) and 2D (left)" with "Figure 1: 3D (left) and 2D (right)"
On page 12, line 174, replace "had been" with "have been"
On page 14, line 211, add a comma between "appeared" and "then"
Author Response
1# On page 2, line 32 replace "analog" with "compound"
Reply: We have corrected as suggested by respected reviewer
2# On page 3, line 58 replace "belong" with "belonging"
Reply: We have corrected as suggested by respected reviewer
3# On page 5, lines 81 and 82, I can't get why you mention the "scheme 1" in the sentence "All synthesized compounds were characterized by different spectroscopic methods given in scheme 1". As a matter of fact, scheme 1 just shows the synthesis of derivatives 1 - 25 and no different spectroscopic methods.
Reply: We have corrected as suggested by respected reviewer
4# On page 9, line 117, replace "flouro" with "fluoro"
Reply: We have corrected as suggested by respected reviewer
5# On page 10, line 137, replace "five" with "four"
Reply: We have corrected as suggested by respected reviewer
6# On page 10, line 139, replace "(Scheme 1)" with "(Table 1)"
Reply: We have corrected as suggested by respected reviewer
7# On page 11, line 154, replace "bending" with "binding"
Reply: We have corrected as suggested by respected reviewer
8# On page 11, line 161, add "of" between "number" and "intermolecular"
Reply: We have corrected as suggested by respected reviewer
9# On page 12, line 169 you have to replace "Figure 1: 3D (right) and 2D (left)" with "Figure 1: 3D (left) and 2D (right)"
Reply: We have corrected as suggested by respected reviewer
10# On page 12, line 174, replace "had been" with "have been"
Reply: We have corrected as suggested by respected reviewer
11# On page 14, line 211, add a comma between "appeared" and "then"
Reply: We have corrected as suggested by respected reviewer
Reviewer 2 Report
The manuscript by Almandil and coworkers reported a series of synthesized quinoxaline analogs as a potential thymidine phosphorylase (TYMP) inhibitor. Among the series, two compounds showed the in vitro enzymatic inhibition with IC50 values around 3.2 μM. The binding of active compounds in the TYMP active site were examined with molecular docking studies.
Although the reported work may be interesting, the authors have already published a series of similar studies on the scaffold of oxadiazole (IC50: 1-50 μM), piperazine (IC50: 0.2 -42 μM), 1,2,4-triazoles (IC50 > 30 μM) and bis-indolylmethane-oxadiazole (IC50; 3.5 μM). It was not clear what the purpose of this work would try to accomplish. Keep adding more scaffolds with similar simple enzymatic inhibition study is just not sufficient enough to move the science forward, which should be highly discouraged for publication. The authors need clearly specify in the manuscript what the specific aim of this study is, what the challenge needs to be solved to develop TYMP inhibitors, and why not further move into cellular and in vivo studies.
Author Response
1# Although the reported work may be interesting, the authors have already published a series of similar studies on the scaffold of oxadiazole (IC50: 1-50 μM), piperazine (IC50: 0.2 -42 μM), 1,2,4-triazoles (IC50 > 30 μM) and bis-indolylmethane-oxadiazole (IC50; 3.5 μM). It was not clear what the purpose of this work would try to accomplish. Keep adding more scaffolds with similar simple enzymatic inhibition study is just not enough to move the science forward, which should be highly discouraged for publication. The authors need clearly specify in the manuscript what the specific aim of this study is, what the challenge needs to be solved to develop TYMP inhibitors, and why not further move into cellular and in vivo studies.
Reply: We agree with the Reviewer that it is a preliminary study. The current project was designed as initial work to identify the potent inhibitors of thymidine phosphorylase. The results of this work encourage further investigation on in vivo studies and more biological tests. We would be happy to carry out the experiment, but we feel sorry for not being in the position to carry out this experiment. The research work was carried out in Atta-ur-Rahman Institute for Natural Product Discovery, Universiti Teknologi Malaysia. The corresponding author has recently relocated to a new position at Department of Clinical Pharmacy, Institute for Research and Medical Consultations (IRMC), University of Dammam, Saudi Arabia, where his group is in its infancy. It will take quite a few months to establish the facilities there.
Reviewer 3 Report
The reviewed paper at title “Synthesis of Thymidine Phosphorylase...” written by Noor Barak Almandil et al. presents a synthesized quinoxaline analogs characterized by 1HNMR... and evaluated for thymidine phosphorylase inhibition. New analogs showed better inhibition when compared with the standard inhibitor. The binding interactions of the active compounds were confirmed through molecular docking studies.
The paper seems to be acceptable but, in my opinion, it requires several modifications. Additionally, several questions should be answered by the authors in detail, as many important issues are described too superficially:
1. In the introduction (start with Line 61), the authors describe the properties of the compounds selected for the study which have nitrogen-containing heterocycles.... However, many others compounds have very similar and attractive properties for example thiadiazoles (e.g. 1,3,4-thiadiazoles) and triazoles exhibit a number of extremely interesting spectroscopic and biological properties, which are also worth mentioning and the recent papers should be cited. For example work: Matwijczuk et al. (especially works about thiadiazoles and fluorescence effects) or Shailee V. Tiwari, Sumaiya Siddiqui et al.
2. Introduction is too short - it should be developed.
3. Lines 112-113 – unmarked figers and not careless. More compounds also - it looks strange.
4. What exactly does “Molecular docking” convey? And why the authors did it?
5. The authors use DMSO, do they realize that DMSO itself can affect the biological aspect of their research?
6. Figure 1 should be corrected. It is too invisible.
7. What is the exact effect of Deazaxanthine (in comparison with the compounds obtained)? Maybe it's worth mentioning at work.
8. Line 593, references from 1979 – are there any more recent studies?
9. Please, complete the references.
10. Conclusions also needs improvement because it is written too superficially.
In conclusion, the paper seems to be acceptable but requires several revisions. The whole layout and neatness of the paper do not leave too much objections, as it is prepared very carefully, but the quality of the discussion requires several amendments.
Please answer all my questions and comments and attach the manuscript with marked changes.
The study itself is rather a description/characterization of the new compounds derived from the relatively quick synthesis and, secondly, a tentative description of the research and calculations performed. For a paper to be published in a reputable journal there is a need for a few sentences presenting an explanation why the study was conducted.
The objections presented by me do not undermine the quality of the paper, which will support in the further publishing process, certainly after careful consideration of my comments.
Author Response
1# In the introduction (start with Line 61), the authors describe the properties of the compounds selected for the study which have nitrogen-containing heterocycles.... However, many others compounds have very similar and attractive properties for example thiadiazoles (e.g. 1,3,4-thiadiazoles) and triazoles exhibit a number of extremely interesting spectroscopic and biological properties, which are also worth mentioning and the recent papers should be cited. For example work: Matwijczuk et al. (especially works about thiadiazoles and fluorescence effects) or Shailee V. Tiwari, Sumaiya Siddiqui et al.
Reply: we have included worthy references as suggested by respected reviewer
2# Introduction is too short - it should be developed.
Reply: We have corrected as suggested by respected reviewer
3# Lines 112-113 – unmarked fingers and not careless. More compounds also - it looks strange.
Reply: We have corrected as suggested by respected reviewer
4. What exactly does “Molecular docking” convey? And why the authors did it?
Reply: Nowadays, computational chemistry is considered as a powerful tool for supporting experimental results. Molecular docking simulation is an important tool in rationalizing and understanding the enzymatic inhibition by drugs and active compounds. Molecular docking can reveal and show how the docked drug or ligand interacts with amino acids into the active site of the targeted enzyme. Thus, here, we use the molecular docking to show how the synthesized compounds interact with the amino acids into the active site of thymidine phosphorylase. In addition, with this tool can explain the higher activity of one compared with another.
5# The authors use DMSO, do they realize that DMSO itself can affect the biological aspect of their research?
Reply: We take the blank of DMSO while calculating effect, so it conceals the effect of DMSO
6# Figure 1 should be corrected. It is too invisible.
Reply: We have corrected as suggested by respected reviewer
7# What is the exact effect of Deazaxanthine (in comparison with the compounds obtained)? Maybe it's worth mentioning at work.
Reply: The structure of our compounds is very close to the standard drug Deazaxanthine but our compounds having fused triazole and thiadiazole ring as well which showed much better activities than standard.
8# Line 593, references from 1979 – are there any more recent studies?
Reply: We have corrected as suggested by respected reviewer
9# Please, complete the references.
Reply: We have corrected as suggested by respected reviewer
10# Conclusions also needs improvement because it is written too superficially.
Reply: We have corrected as suggested by respected reviewer
Round 2
Reviewer 2 Report
As indicated in the review comment earlier, the scientific significance is just not there. Thus, it is not recommended for publication.
Author Response
Dear reviewer we have provided the rational as suggested but regarding in your last comments you mentioned that we published similar studies please make it clear that although we have screened with same enzyme but all series having different scaffold and there synthesis are also totally different. editor can also see this.